# Harnessing the Loop: The Perspective of Circular RNA in Modern Therapeutics

**DOI:** 10.3390/vaccines13080821

**Published:** 2025-07-31

**Authors:** Yang-Yang Zhao, Fu-Ming Zhu, Yong-Juan Zhang, Huanhuan Y. Wei

**Affiliations:** Shanghai Institute of Nutrition and Health, University of Chinese Academy of Sciences, Chinese Academy of Sciences, Shanghai 200031, China; zhaoyangyang2024@sinh.ac.cn (Y.-Y.Z.); zhufuming2023@sinh.ac.cn (F.-M.Z.)

**Keywords:** circRNA vaccines, circRNA-based therapeutics, immune response of circRNA, design strategies, AI enablement

## Abstract

Circular RNAs (circRNAs) have emerged as a transformative class of RNA therapeutics, distinguished by their closed-loop structure conferring nuclease resistance, reduced immunogenicity, and sustained translational activity. While challenges in pharmacokinetic control and manufacturing standardization require resolution, emerging synergies between computational design tools and modular delivery platforms are accelerating clinical translation. In this review, we synthesize recent advances in circRNA therapeutics, with a focused analysis of their stability and immunogenic properties in vaccine and drug development. Notably, key synthesis strategies, delivery platforms, and AI-driven optimization methods enabling scalable production are discussed. Moreover, we summarize preclinical and emerging clinical studies that underscore the potential of circRNA in vaccine development and protein replacement therapies. As both a promising expression vehicle and programmable regulatory molecule, circRNA represents a versatile platform poised to advance next-generation biologics and precision medicine.

## 1. Introduction

The history of humanity’s pursuit of health and longevity is a chronicle of battles against diseases. Conventional vaccine modalities, encompassing inactivated vaccines, live attenuated vaccines, subunit vaccines, and recombinant protein vaccines [1], elicit immune responses by preserving antigenic structures to confer protection. However, these approaches are characterized by prolonged production timelines, substantial research and development expenditures, and inherent safety concerns [2]. The advent of the COVID-19 pandemic catalyzed the rapid deployment of mRNA vaccines, notably mRNA-1273 [3] and BNT162b2 [4], which demonstrated rapid adaptability and enhanced safety profiles, emerging as the first mRNA-based vaccines to achieve clinical authorization.

mRNA vaccines function by introducing synthetic mRNA encoding viral antigens to prime the immune system for antibody production. By leveraging the central dogma of molecular biology, this innovative paradigm harnesses the host cellular machinery for endogenous protein synthesis, compressing traditional vaccine development cycles—typically spanning over a decade—to a mere 11 months during the pandemic. In the post-pandemic era, mRNA-based therapeutic applications continue to evolve, extending beyond prophylactic vaccination to include other therapeutic applications. Theoretically, mRNA platforms enable the synthesis of diverse functional proteins or peptides for applications in vaccination, protein replacement therapy, or disease modulation [5]. Among emerging mRNA-derived modalities, circular RNA (circRNA)-based therapeutics have garnered significant attention. These utilize synthetic, covalently closed circRNA molecules as expression vectors to encode therapeutic proteins. CircRNA vaccines exhibit superior stability, reduced immunogenicity, sustained antigen expression kinetics, rapid scalability, and adaptability to pathogen mutations, positioning them as a transformative advancement in next-generation mRNA therapies [5,6]. The low immunogenicity of circRNA obviates the necessity for expensive modifications such as pseudouridine and methylpseudouridine incorporation, which is essential for attenuating innate immune responses in linear mRNA constructs.

Furthermore, their enhanced thermodynamic stability reduces cold-chain dependency, thereby lowering logistical costs and enhancing suitability for industrial-scale production. Preliminary advancements in circRNA-based pharmaceutical development have been reported globally [6]. Nevertheless, as a nascent technology, circRNA vaccines and therapeutics necessitate extensive optimization across sequence design, manufacturing protocols, delivery systems, and clinical validation prior to widespread implementation.

CircRNA, a covalently closed RNA species, was first identified in plant viroids by Sanger et al. in 1976 [7]. Subsequent studies have detected circRNA in HeLa cells [8], yeast [9], and hepatitis delta virus (HDV) [10], although these molecules were initially dismissed as transcriptional byproducts. The advent of next-generation sequencing in 2012 revealed the ubiquitous presence of circRNA in mammalian cells [11,12], with functional roles as competitive endogenous RNAs (ceRNAs) that sequester microRNAs to regulate gene expression [13]. circRNAs were previously classified as non-coding despite early evidence of exogenous circRNA translation in cell-free systems [14]. A paradigm shift occurred in 2015 when the Wang Zefeng team at UNC demonstrated the translation of exogenous circRNA in mammalian cells [15], followed by seminal studies in 2017 that confirmed the translation of endogenous circRNA [16,17,18]. These discoveries reignited scientific interest in the translation of circRNA.

Although numerous endogenous circRNAs have been identified, large-scale in vitro synthesis of circRNA has been a challenge. Earlier methodologies, such as T4 RNA ligase-mediated circularization with DNA splints, faced limitations in scalability and purification [14,19,20]. A pivotal milestone in the therapeutic application of circRNA was achieved in 2018 by the Daniel Anderson team at MIT [21]. By leveraging Group I self-splicing introns, they optimized the permuted introns and exons (PIE) system [22], enabling the generation of linear RNA constructs that undergo efficient in vitro circularization while retaining the capacity to express target proteins.

In 2022, Wang Zefeng’s team at CAS advanced this field by developing a Group II self-splicing intron-based PIE methodology, facilitating scarless circRNA production under mild reaction conditions. These innovations have accelerated the synthesis of circRNA toward industrial feasibility. By late 2024 to early 2025, Chinese biotechnology firms Ribox Pharmaceuticals and CirCode Biotech secured Investigational New Drug (IND) approvals from the U.S. FDA and China’s NMPA, underscoring the translational potential of circRNA-based therapies (clinicaltrials.gov).

Despite the evident advantages of circRNA vaccines, their developmental phase remains nascent, with persistent challenges in sequence optimization, scalable manufacturing, targeted delivery, and rigorous in vivo validation. This review synthesizes recent advancements in circRNA vaccine design, production methodologies, delivery platforms, immunogenicity profiling, preclinical evaluations, and clinical trials, with a focus on their stability and degradation, immunogenic properties in vaccines and drugs, and AI-driven innovations in circRNA therapeutics. The objective is to provide a comprehensive reference that catalyzes further research and expedites the transition of circRNA vaccines/medicines from foundational discoveries to clinical applications.

## 2. Characteristics of circRNA Therapeutics

### 2.1. Fundamental Properties of circRNA

CircRNAs are primarily generated through back-splicing of precursor mRNAs, where the 3′ terminus of an upstream exon covalently links to the 5′ terminus of a downstream exon, forming a stable closed-loop structure devoid of 5′ caps or 3′ poly(A) tails. Based on their composition and origin, circRNAs are classified into three categories: exonic circRNAs (ecircRNAs, comprising single or multiple exons), exon–intron circRNAs (elciRNAs), and intronic circRNAs (ciRNAs) [23,24] (Figure 1A). CircRNAs are abundant, evolutionarily conserved, and exhibit cell- or tissue-specific expression patterns [25,26,27,28]. They perform diverse biological functions, including acting as miRNA sponges [13,29], interacting with mRNAs [30,31,32] and proteins [33,34] to regulate transcription and protein expression. They can form functional circRNP complexes to modulate signaling pathways [35] and encode proteins [15,16,36,37,38]. Comprehensive reviews of these functions are available [25,39,40,41,42].

Despite their diverse endogenous roles, the potential of circRNAs as therapeutic agents hinges on overcoming the challenge of efficient and controlled in vitro production. Current in vitro synthesis strategies include chemical synthesis, ligase-based circularization, and PIE strategy. Chemical synthesis, hampered by multi-step processes, harsh conditions, and low efficiency, is not discussed here. Ligase methods employ enzymes like T4 DNA ligase, T4 RNA ligase 1 (Rnl1), and T4 RNA ligase 2 (Rnl2) to circularize single-stranded RNA (ssRNA), though they face challenges with sequence constraints and concatemer formation (Figure 1B, upper panel). The PIE strategy is currently the most valuable method due to its high efficiency and precision. It utilizes self-splicing Group I or Group II introns as ribozymes to catalyze back-splicing. Group I circularization requires GTP and Mg^2+^ cofactors. Group II induces exon circularization via a two-step transesterification reaction without introducing GTP, achieving high-efficient back-splicing in vitro (Figure 1B, lower panel).

Compared to linear RNA, circRNAs exhibit superior stability, reduced immunogenicity, and prolonged protein expression, positioning them as a promising novel platform for RNA-based therapeutics.

### 2.2. Stability and Degradation Mechanisms of circRNA

CircRNA vaccines possess superior stability relative to linear mRNA vaccines. Cellular degradation of linear mRNA predominantly relies on deadenylation, which removes the 5′ cap and facilitates 5′-to-3′ exonuclease-mediated decay [43,44]. In contrast, circRNAs lack terminal structures, rendering them resistant to exonucleases and thus inherently more stable. Although evading classical RNA degradation pathways, several degradation mechanisms of circRNAs have been identified in both the nucleus and cytoplasm (Figure 2): (1) **Argonaute 2 (Ago2)** participates in microRNA-mediated circRNA degradation [45,46]. (2) **RNase H1** degrades circRNAs by recognizing RNA-DNA hybrid R-loops [47]. (3) **RNase L**, activated during viral infection or inflammation, mediates rapid degradation of circRNAs bound to protein kinase R (PKR) [48]. (4) Recently, Murakami and colleagues found m^6^A-modified RNAs could induce **ribosome stalling and collision,** like mRNAs, which recruits YTHDFs to initiate degradation [49]. (5) m^6^A-modified circRNAs are directly recognized by YTHDF2 and degraded via **HRSP12-RNase P/MRP complexes** [50]. (6) **Structure-mediated RNA degradation (SRD)** involves UPF1 and G3BP1 in degrading highly structured circRNAs [51,52]. (7) In February 2025, a collaborative study by Ling-Ling Chen and Li Yang identified a **DIS3-dependent circRNA degradation pathway** under physiological conditions, offering insights for engineering degradation-resistant circRNAs to enhance vaccine stability [53]. Compared to broad inhibition of RNA degradation pathways (e.g., RNase L or Ago2), targeted suppression of DIS3 exhibits higher specificity. DIS3 primarily participates in ribosomal RNA processing and mRNA degradation, yet its role in circRNA metabolism may be functionally distinct. Selective inhibition of DIS3-mediated circRNA degradation activity minimizes interference with cellular RNA homeostasis, thereby reducing the risk of off-target effects. By inhibiting DIS3 activity (e.g., through small-molecule inhibitors or gene editing technologies), the cleavage of circRNA can be directly blocked, thereby prolonging its half-life.

Notably, DIS3 prefers to degrade circRNAs containing U-rich motifs under normal physiological conditions. However, when cells are infected by viruses, RNase L is activated and globally degrades circRNAs, and the DIS3 pathway is suppressed. This dynamic switching reveals a competition between the dominant degradation pathways under pathological versus physiological states, suggesting complex interactions may exist between distinct degradation mechanisms. Differently, in colorectal cancer, tumor cells actively excrete tumor-suppressive circRNAs, such as circRHOBTB3, out of the cell via exosomes to eliminate these circRNAs and thereby maintain oncogenic fitness (tumor-suppressive circRHOBTB3 is excreted out of cells via exosome to sustain colorectal cancer cell fitness, [54]). This indicates that circRNA degradation pathways may exhibit specificity across different cell types and diseases. A comprehensive understanding of circRNA degradation mechanisms necessitates further research.

In summary, although circRNA is more stable compared to linear mRNA, understanding its degradation mechanisms provides a deeper insight into its lifecycle, which could enhance its enhance its stability and extend its therapeutic window to a greater extent.

### 2.3. Immune Responses to circRNA

A hallmark of highly purified circRNAs is their low immunogenicity compared to linear RNAs, which is primarily attributed to the rigorous removal of contaminants during purification. Crucially, impurities such as residual linear RNA fragments, double-stranded RNA (dsRNA), un-capped precursor RNA, and protein impurities (e.g., host cell proteins or endotoxins) are potent activators of innate immune sensors like RIG-I. The presence of these contaminants in insufficiently purified preparations is a major reason for the seemingly contradictory findings reported in the literature. Therefore, the effective removal of these contaminants significantly reduces innate-immunity-mediated degradation and facilitates prolonged expression. Early studies suggested that exogenous circRNAs have higher immunogenicity, which can activate the RIG-I pathway and induce innate immunity to suppress viral infections [55]. However, recent reports have shown that circRNAs can essentially bypass the innate cellular immune system, such as the RIG-I-mediated pathway, when carefully purified to eliminate dsRNA and uncapped precursor RNA [56,57,58]. In summary, the conflicting conclusions reported in prior studies are largely attributable to uncontrolled contamination events occurring at the purification stage. Therefore, advances in circRNA purification methodologies represent a pivotal frontier for generating reproducible datasets in this field.

The immunogenicity of circRNAs is a complex issue. Key factors influencing their immunogenicity include the following:(1)Sequence and structure: dsRNA regions, GU/U-rich motifs, and imperfect circularization enhance immunogenicity.(2)Cellular context: Immune cell type (e.g., dendritic cells vs. epithelial cells), subcellular localization, and RNA abundance dictate response magnitude.(3)Delivery method: Exogenous circRNAs (e.g., synthetic or encapsulated) are more immunogenic than endogenous circRNAs. When circRNAs enter cells (e.g., via exogenous delivery or endogenous release), they can activate immune pathways through multiple mechanisms.

Notably, the immunogenicity of circRNA is context-dependent. For instance, it was found to be affected by the exact RNA sequence, the method of synthesis, and the type of base modification (Figure 3).

#### 2.3.1. Recognition by Cytosolic Immune Sensors

CircRNAs are primarily detected by pattern recognition receptors (PRRs) in the cytoplasm, which recognize conserved molecular patterns associated with pathogens or cellular damage. In the RIG-I-like receptors (RLRs), RIG-I (Retinoic acid-inducible gene-I) and MDA5 (Melanoma differentiation-associated protein 5) are key sensors for viral RNA. CircRNAs with double-stranded regions or unpaired termini (e.g., resulting from imperfect circularization) can mimic viral RNA structures, thereby activating RIG-I/MDA5. The activation triggers MAVS (mitochondrial antiviral-signaling protein)-dependent signaling, leading to the production of type I interferon (IFN-α/β) and pro-inflammatory cytokines (e.g., IL-6, TNF-α).

The PRRs also include the PKR. PKR binds to double-stranded RNA (dsRNA) motifs in circRNAs. Prolonged PKR activation induces the phosphorylation of eIF2α, halting global translation (an antiviral response) and promoting the formation of stress granules. Some circRNAs sequester PKR, paradoxically inhibiting its activity and dampening immune responses [48]. On another aspect, circRNA binding to PKR initially inhibits its activity, followed by RNase L-mediated degradation, which releases PKR to activate innate antiviral responses [48,59,60].

#### 2.3.2. Endosomal Toll-like Receptor (TLR) Activation

Exogenous circRNAs (e.g., delivered via lipid nanoparticles or extracellular vesicles) may enter endosomes, where they engage TLR3, TLR7, or TLR8. TLR3 recognizes dsRNA, while TLR7/8 sense single-stranded RNA (ssRNA). TLR activation recruits adaptors (e.g., MyD88, TRIF), driving NF-κB and IRF3/7 pathways to produce inflammatory cytokines and IFNs [61].

#### 2.3.3. Inflammasome Activation

CircRNAs can indirectly activate the NLRP3 inflammasome. PKR or RLR signaling increases reactive oxygen species (ROS) and potassium efflux, priming NLRP3. Inflammasome assembly cleaves pro-caspase-1 to active caspase-1, which processes IL-1β and IL-18 into their mature, secreted forms [62].

#### 2.3.4. Adaptive Immune Responses

CircRNAs may also influence adaptive immunity. The peptides generated by circRNAs may be presented on MHC-I, activating CD8+ T cells. Meanwhile, circRNAs with repetitive motifs (e.g., Alu elements) could crosslink B cell receptors (BCRs), promoting antibody production [63].

#### 2.3.5. Immune Evasion Strategies

Some circRNAs suppress immune activation, thereby preventing excessive inflammation.

(1)Sponging immune mediators: circRNAs sequester miRNAs (e.g., miR-7) or proteins (e.g., PKR) that regulate immune signaling [13,48].(2)Masking immunostimulatory motifs: Perfectly circularized RNAs lack free ends, reducing RIG-I recognition [56].

#### 2.3.6. Immune Activation as Adjuvants

When combined with soluble proteins, exogenous circRNA can serve as excellent vaccine adjuvants, inducing strong T cell responses in multiple immune pathways [64,65].

### 2.4. Synthesis Strategies, Quality Control, and Delivery Methods of circRNA Vaccines

As covalently closed circular structures, the circularization efficiency of circRNA vaccines directly impacts their translation efficiency and stability. Enzymatic ligation methods exhibit limited circularization efficiency, whereas PIE systems can significantly enhance circularization rates [21].

However, challenges such as increasing circularization efficiency, reducing costs in large-scale production processes, minimizing exogenous sequences, and improving efficient purification persist. Breakthroughs in manufacturing processes rely on innovations in circularization technologies and the establishment of scalable production platforms. Group II intron-based self-splicing system achieves high-efficiency RNA circularization through two-step transesterification reactions, without introducing “scar” sequences [58]. Notably, CureMed disclosed a novel Clean-PIE methodology in 2022, which employs systematic screening of protein-coding regions or internal ribosome entry site (IRES) elements to identify optimal circularization loci, thereby enabling exogenous sequence-free RNA cyclization [66]. Similarly, enhanced chimeric permuted intron–exon (CPIE) systems, derived from Group I introns, efficiently circularize linear mRNA molecules while reducing potential immunogenicity [67]. Notably, the MRC Laboratory of Molecular Biology (MRC-LMB) developed a trans-acting ribozyme-based circularization (TRIC) approach utilizing Anabaena tRNA-derived type I introns. This innovative strategy utilizes the internal guide sequence (IGS) to mediate spatial approximation between the P1 and P0 helices through interactions with flanking exons, thereby enabling the precise covalent linkage of target sequences to the 3′-terminus of trans-acting ribozymes for practical synthesis of circRNA [36].

Notably, residual linear RNA impurities may trigger immune responses or compromise vaccine efficacy [56]. Thus, purification methods such as high-performance liquid chromatography (HPLC) or gel electrophoresis are crucial for removing uncircularized RNA. Additionally, contaminants such as DNA templates, enzymes, and chemical reagents must be removed to ensure vaccine purity and safety. Additionally, the stability of the circRNA vaccine needs to be monitored and verified in real time, which is crucial for the development of the vaccine. Quality control during production should focus on circularization efficiency, purity, stability, and immunogenicity. Advanced purification techniques, including preparative HPLC, ion-exchange chromatography (IEX), and size-exclusion chromatography (SEC), exploit differences in polarity or molecular size between circRNA and linear RNA. Designing engineering circRNA with affinity tags (e.g., customized tags, biotin, His-tags, or aptamers) enables targeted purification through ligand-specific binding. A recent study demonstrated that ultrafiltration-based purification of circRNA from in vitro transcription (IVT) products achieves twice the purity of size-exclusion high-performance liquid chromatography (SE-HPLC) methods, with a yield exceeding 50%, highlighting its potential to enhance the accessibility of circRNA vaccines [68].

Following purification, engineered circRNA vaccines can be administered via multiple delivery platforms. Current mainstream strategies include lipid nanoparticle delivery systems (LNP), direct injection strategies for naked circRNA vaccines, and ultrasound-guided percutaneous injections. Notably, CureMed synthesized a series of ionizable lipids with multiple ester bonds in the branch tail through a Michael addition reaction, with structural validation by ^1^H NMR spectroscopy, and finally screened AX4-LNP [69]. This optimized reagent demonstrated efficient hepatic delivery of circRNA cmRNA-1130, characterized by rapid degradation kinetics and low cytotoxicity in hepatocytes, thereby highlighting its translational potential. Also, Jia et al. engineered the circSCMH1@LNP1 complex for intranasal administration of circSCMH1 to ameliorate post-stroke cerebral injury [70]. Furthermore, Yang et al. achieved direct tumor cell transfection by injecting naked luciferase-encoding cmRNA dissolved in phosphate-buffered saline (PBS), with detectable luminescence within 6 h post-injection, confirming endocytosis-mediated cellular uptake and functional protein expression [71]. For comprehensive insights into circRNA vaccine delivery strategies, readers are directed to specialized review articles [2,23,43,72,73].

Since 2020, the Center for Drug Evaluation of China’s National Medical Products Administration (CDE.NMPA) [74], the World Health Organization (WHO) [75], and United States Pharmacopeia (USP) [76] have issued a series of guidelines on mRNA vaccines to evaluate their quality, safety, and efficacy in preventing infectious diseases, as well as to regulate the production and control of mRNA vaccines and their components. Currently, international standards for circRNA vaccines remain under development. Moreover, guidelines for nucleic acid vaccine, such as mRNA vaccines, can serve as interim references for quality control during the production of circRNA vaccines.

## 3. Sequence Design and Optimization Strategies for circRNA Therapeutics

Optimization of the nucleic acid design framework in circRNA synthesis significantly enhances the yield of circRNA and protein production [37]. Since PIE strategies are the most popular methods applied in circRNA production in manufacturing, owing to their demonstrable advantages in scalability and yield for multi-kilobase constructs over enzymatic alternatives like T4 DNA ligase, we focus on sequence design and optimization strategies based on this approach despite its known sequence-dependent efficiency constraints. Specifically, the nucleic acid design framework based on the PIE methods comprises the following key components, as illustrated in Figure 4: (1) Gene of Interest (GOI), which encodes the target protein. (2) IRES to initiate circRNA translation. (3) Group II intron that is responsible for self-splicing-mediated circularization. Despite the essential element, some auxiliary elements facilitate circularization: (4) spacer sequences to prevent structural interference between the Group II intron and IRES, and (5) exogenous base-paired homologous arms to spatially facilitate circularization.

Design and optimization strategies focus on two critical dimensions: splicing capacity and translational efficiency. Translational optimization includes the following: (a) Systematic screening and synthesis of high-efficiency IRES elements: leverage bioinformatics tools (e.g., CIRI2, circRNAprofiler, circtools), circRNA databases (circBank, circBase, circAtlas, PlantCircNet), and IRES databases (IRESite, IRESbase) to identify or engineer natural/synthetic IRES with enhanced activity. (b) Codon optimization of the GOI: tailors sequences to host-specific codon usage biases using quantum computing, recurrent neural networks (RNNs), or deep learning (DL). (c) Incorporation and refinement of 5′ and 3′ untranslated regions (UTRs). Splicing optimization includes (d) Design, screening, and optimization of Group I/II intron sequences. (e) Systematic refinement of spacer sequences to minimize steric hindrance. (f) Design of exogenous homologous arms and their compatibility optimization with other components (primarily introns).

### 3.1. Optimization of Open Reading Frame Sequences

The design and optimization of circRNA vaccines require meticulous selection of open reading frame (ORF) sequences with tailored attributes, including antigenicity, immunogenicity, compatibility with IRES, stability, and translational efficiency. A primary focus is identifying viral or tumor-derived sequences with broad immunogenic potential. Combinatorial libraries serve as powerful tools for discovering cross-reactive epitopes. In contrast, bioinformatics tools such as BepiPred-2.0 (for B cell epitope prediction) and NetMHCpan-4.0 (for T cell epitope and neoantigen identification) enable systematic screening of immunogenic sequences. These tools target immune evasion mechanisms employed by pathogens and tumors, such as reducing the affinity between antigens and human leukocyte antigen (HLA) class I. Population-level HLA allele frequency analysis further ensures the applicability of vaccines across diverse demographics.

#### 3.1.1. Codon Usage Bias (CUB) and Optimization

Codon usage bias, shaped by genomic GC content, tRNA abundance, and evolutionary pressures, profoundly impacts mRNA stability, translational efficiency, and co-translational protein folding. Codon optimization replaces rare codons with host-preferred synonymous codons to enhance heterologous protein expression. For instance, codon-optimized mRNA for SARS-CoV-2 Omicron variants has demonstrated elevated protein yields in experimental models. Advanced algorithms now integrate multi-dimensional approaches:(1)Quantum computing optimizes GC content and minimizes nucleotide repeats.(2)DL models, such as iDRO (integrated deep-learning-based mRNA optimization) [77] and mRNA-LM (language model) [78] predict full-length mRNA sequences for maximal expression.(3)RNNs, including bidirectional LSTM architectures, learn codon usage patterns to recommend optimal codon substitutions (e.g., the ICOR tool for *E. coli*) [79].(4)Mixed-integer linear programming balances codon selection with secondary structure constraints [80].

#### 3.1.2. Integrated Algorithmic Platforms

Traditional optimization strategies often focus on singular objectives, such as codon bias or thermodynamic stability, neglecting their interdependencies. To address this, the LinearDesign algorithm jointly optimizes mRNA sequences for stability and codon usage, yielding enhanced vaccine efficacy [81]. Building on this, the circDesign platform [66,82] extends these principles to circRNA in the following ways: (1) Generating candidate ORFs using LinearDesign. (2) Assembling these ORFs with IRES motifs and functional elements. (3) Evaluating positional entropy and structural diversity to predict circRNA performance. (4) Mitigating cross-interactions between ORFs and IRES motifs to preserve IRES functionality. Comparative studies demonstrate that circDesign-engineered circRNAs exhibit superior circularization efficiency, stability, protein expression, and immune activation compared to conventionally optimized sequences. This platform also minimizes interference from residual exonic fragments during RNA folding, ensuring robust vaccine design.

Emerging technologies, such as AI-driven epitope prediction and dynamic codon optimization frameworks, hold promise for further refining the design of circRNA vaccines. Integrating multi-omics data and in silico modelling will enable the development of personalized vaccines tailored to individual HLA profiles and pathogen evolution.

### 3.2. Selection and Optimization of IRESes

circRNAs can drive cap-independent protein translation through IRES. IRES elements are structured sequences located upstream of the initiation codons that recruit the 40S ribosomal subunit, enabling translation initiation under stress or during viral infection [83,84]. Mass spectrometry studies confirm that IRES-containing circRNAs produce detectable proteins, validating their translational capacity [16,85].

Even small fragments can promote the translation of circRNAs as IRES-like elements. Moreover, the final translation efficiency of circRNAs is closely related to IRES [38]. IRES-mediated translation efficiency varies across different cellular contexts, necessitating a flexible selection approach based on the specific application [21]. Therefore, selecting and optimizing IRES elements is crucial for the development of circRNA-based therapeutics. IRES selection, databases, resources, and IRES-ORF integration are aspects that need to be considered.

#### 3.2.1. IRES Databases and Resource Platforms

IRESdb: Catalogues published IRES sequences with functional annotations [86].

IRESit: Provides detailed biological metadata (e.g., cellular context, activity levels) for 92 parameters, aiding IRES screening and design [87].

IRESbase: A curated repository of 1328 experimentally validated minimal IRES elements (774 eukaryotic, 554 viral), averaging 174 nucleotides in length. This resource supports circRNA engineering by focusing on essential functional motifs [88].

#### 3.2.2. Design Considerations for IRES-ORF Integration

The IRES-ORF integration is important for translation outcome [37]. Key design principles include the following: (1) Length Optimization: Shorter circRNAs reduce secondary structure complexity and looping stress. Kozak sequences or AU-rich motifs are preferred for compact IRES design [89,90]. (2) Immunogenic Prioritization: MHC ligand sequencing identifies shorter antigenic epitopes, enabling replacement of lengthy ORFs with immunodominant sequences [91]. (3) Spacer Integration: Inserting spacers between IRES and splicing junctions minimizes interference, enhancing circularization efficiency [21]. Besides IRES-ORF integration, engineered circRNAs often incorporate UTR-like sequences flanking IRES-ORF cassettes to mimic RNA-binding protein (RBP) interaction sites to promote translation efficiency [2].

### 3.3. Optimization Strategies for Group I or II INTRONS and Other Components to Enhance Splicing Efficiency

#### 3.3.1. Design and Optimization of Intronic Ribozymes with High Circularization Efficiency

To develop intronic ribozymes with enhanced circularization efficiency, a systematic approach that integrates experimental and computational strategies is essential. The process begins with the collection and classification of intron sequences from public databases, prioritizing diversity across species (bacteria, archaea, and lower eukaryotes) and subtypes (IIA, IIB, IIC). Functional validation of these introns for self-splicing activity is crucial, particularly in terms of their catalytic core domains. For instance, Domain V (D5), the catalytic core, can be optimized using in vitro evolution techniques such as random mutagenesis coupled with high-throughput sequencing and activity screening (e.g., SELEX). This step aims to enhance splicing efficiency while preserving structural integrity.

Parallel efforts include exploring trans-splicing strategies to improve RNA circularization yields [36]. Unlike canonical cis-splicing, trans-splicing retains intronic secondary structures, which may enhance catalytic activity. Structural–functional relationships are further analyzed using tools such as RNAfold and mFold to predict how features like stem-loops and pseudoknots influence ribozyme performance.

#### 3.3.2. Optimization of Spacer Sequences in the PIE System

The role of spacer sequences in the PIE framework could maximize circRNA yields. Systematic experiments could be used to assess whether spacer incorporation improves efficiency. Subsequent optimization should focus on spacer length, GC content, and physicochemical properties (e.g., hydrophilicity, secondary structure) using computational tools such as EMBOSS and Galaxy. Empirical validation identifies optimal spacer parameters, ensuring compatibility with the PIE system.

#### 3.3.3. Design and Matching of Homology Arms in the CirCode System

Homology arm design has a significant impact on splicing efficiency. Studies evaluate how arm length and GC content influence circularization rates, using high-throughput screening of oligonucleotide libraries and barcoding techniques to identify optimal configurations. Orthogonal experiments further validate synergistic effects between homology arms and other system components, refining the overall design.

This integrated approach—combining experimental validation, AI-driven modelling, and systematic optimization—provides a robust pipeline for advancing RNA circularization technologies, with broad applications in synthetic biology and therapeutic development.

### 3.4. Design Strategies to Enhance the Stability of circRNA

The design strategies to enhance the stability of circRNA mainly depend on their resistance to degradation.

(1)Avoid incorporating degradation signals, such as specific nucleotide sequences or secondary structures.(2)Avoid introducing high burden of methylation (e.g., m^6^A) or other chemical modifications to initiate circRNA degradation.(3)Engineer hairpin structures or protein-binding motifs into circRNA to generate steric hindrance, thereby blocking degradation by enzymes that could cleave circRNA (e.g., DIS3).

## 4. Artificial Intelligence (AI) in Advancing circRNA Vaccines

### 4.1. The Rational Design

Advancements in artificial intelligence (AI) offer transformative opportunities for rational ribozyme design. A key initiative involves building an open-access database of intronic activity data, integrating sequence features, structural motifs, and environmental parameters. Machine learning models, such as convolutional neural networks (CNNs) for capturing local sequence patterns (e.g., conserved motifs) and graph neural networks (GNNs) for processing secondary structure graphs, are trained to predict catalytic activity. Model interpretability and iterative validation with experimental data ensure biological relevance, enabling the design of highly efficient ribozymes.

The rational design and optimization of circRNA sequences to enhance their stability and translational capacity as vaccine platforms present significant challenges. To address this, researchers have developed CircDesign, an AI-driven algorithm that facilitates the design of circRNA sequences exhibiting enhanced circularization efficiency, improved stability, and increased translational efficiency. The accuracy of this algorithm also has been empirically validated using the rabies virus glycoprotein (RABV-G) and varicella-zoster virus (VZV) glycoprotein gE as model antigens [92].

### 4.2. Model Construction and Databases

Furthermore, researchers have constructed the metaCDA model, which utilizes meta-networks and adaptive aggregation systems for the effective prediction of disease-associated circRNAs [93]. Concurrently, through algorithm optimization and multi-omics data integration, Ma Hu’s team developed the TCCIA database, a comprehensive resource for circRNAs in cancer immunotherapy [94]. The establishment of circRNA databases provides a critical foundation for leveraging AI to advance circRNA vaccine development.

### 4.3. Manufacturing

Additionally, AI facilitates large-scale manufacturing of circRNA vaccines. For instance, China National Biotec Group (CNBG) has integrated computer vision technology with existing process monitoring systems, leveraging AI to enable centralized, real-time oversight of critical biosafety protocols and aseptic operations throughout vaccine production. AI is thus poised to become a cornerstone in enhancing production safety and elevating quality control standards for circRNA vaccines upon regulatory approval and transition to mass manufacturing.

### 4.4. Clinical Translation

In the clinical translation of circRNA vaccines, AI significantly accelerates the development of personalized vaccines by facilitating dynamic antigen expression adjustments and tailoring dosing regimens based on patient-specific data. The integration of multi-omics data with in silico predictive models enables the design of precision therapies tailored to individual HLA diversity and evolving pathogen dynamics. Currently, AI has been applied to antigen design. For example, Chu et al. developed TransPHLA, a transformer model-based program that autonomously optimizes mutant peptides with enhanced binding affinity to HLA alleles and high homology to source peptides, thereby accelerating immunogenic peptide screening and advancing vaccine design [95]. Similarly, Gulam Sarwar et al. established ImmueMirror, a machine learning-powered web-based platform for antigenic epitope prediction [96].

Recent work by Marvin E. Tanenbaum’s team has pioneered a high-resolution dynamic imaging technology for monitoring single-ribosome translation within monosomes or polysomes, based on computational modelling and analysis of stop codon-deficient circRNAs (socRNAs) [97]. SocRNAs, as circRNAs lacking in-frame termination codons, permit precise quantification of translation elongation kinetics. Leveraging this principle, researchers generated socRNAs in cells via the Tornado system and visualized their dynamics using the SunTag translation imaging platform. However, frequent collisions between ribosomes translating the same socRNA molecules were observed. To quantify ribosome collision frequencies, the team implemented stochastic computer simulations of translation elongation dynamics. Interestingly, transient ribosome collisions exhibited a median duration of ~3 min, with collision patterns remaining consistent across variable simulation parameters. Collectively, the socRNA platform integrates computational simulations with live imaging to enable long-term observation of single-ribosome translation trajectories, unveiling cooperative mechanisms underlying transient ribosomal collisions. This advancement underscores the transformative potential of computer simulation in elucidating the functional dynamics of circRNAs.

Looking ahead, as AI transitions from a supportive tool to a foundational driver in circRNA vaccine development, its core value extends beyond accelerating iterative design optimization (Figure 5). Crucially, AI enables the exploration of previously inaccessible biological design spaces—such as engineering ultra-stable circRNA or high-efficiency translational circRNA constructs—that transcend conventional experimental approaches. Nevertheless, AI-driven circRNA vaccines research remains in an early developmental phase, with significant challenges to overcome. The deepening synergy between AI algorithms and synthetic biology promises transformative breakthroughs in circRNA vaccine applications, particularly for infectious disease prophylaxis and cancer immunotherapeutics. Beyond vaccines, AI concurrently expedites the advancement of circRNA-based therapeutics, thereby establishing circRNA as a programmable therapeutic platform.

## 5. Recent Advances in circRNA Therapeutics

CircRNA vaccines, characterized by high stability, low immunogenicity, and sustained antigen expression, elicit potent immune responses—including T cell and B cell activation—in vivo [2,64]. Comparatively, circRNA vaccines confer several advantages over mRNA vaccines and other traditional vaccines (summarized in Table 1) [2,98,99,100,101,102,103,104]. [reference: https://doi.org/10.3390/vaccines4020012; https://doi.org/10.1016/j.tvjl.2017.12.025; https://doi.org/10.1128/cmr.00241-21; https://doi.org/10.3389/fimmu.2019.00594; https://doi.org/10.1016/j.addr.2020.12.011; https://doi.org/10.1038/s41392-023-01561-x; https://doi.org/10.1080/22221751.2024.2396886; https://doi.org/10.3390/pathogens13080692]. These attributes underscore their therapeutic potential in preventing and treating viral infections, cancers, autoimmune disorders, and metabolic diseases.

CircRNA vaccines represent a transformative advance over mRNA platforms, critically addressing their core limitation of thermolability. This is exemplified by studies demonstrating exceptional stability: for instance, engineered small circRNA vaccines (<300 nt) exhibit a half-life of 400 days at −20 °C, drastically exceeding the days-to-weeks stability of conventional mRNA vaccines (small circular RNAs as vaccines for cancer immunotherapy, DOI: 10.1038/s41551-025-01344-5, Reference [59]). Such resilience significantly reduces dependence on ultra-cold chain logistics (e.g., −70 °C for mRNA), enabling storage at readily accessible temperatures (e.g., 4 °C or even ambient in optimized formulations). Beyond thermostability, circRNA vaccines elicit robust and durable immunogenicity against pathogens like influenza and Zika [105,106,107], attributed to sustained antigen expression that potentiates long-lasting immune responses, often outperforming the transient expression profile of mRNA. Combined with rapid design adaptability for emerging variants, these properties position circRNA as a potent platform for pandemic response and resource-limited settings. To fully realize this potential, ongoing development must prioritize (1) sequence optimization for enhanced circularization/expression [38], (2) further reduction of innate immunogenicity [65], (3) targeted delivery systems [69], and (4) scalable, cost-effective manufacturing.

Beyond infectious disease control, circRNA vaccines are poised to play pivotal roles in cancer immunotherapy, autoimmune disease management, and metabolic disorder treatment. For instance, circRNAs engineered to encode tumor-specific antigens [108] can induce targeted immune responses against malignancies. As circRNA technology matures, it is expected to usher in a new era of drug development, laying the foundation for next-generation therapeutic paradigms. Researchers have explored their potential applications in diverse innovative therapeutic strategies, including vaccines for infectious diseases [105,107,109], cancer vaccines [108,110], in vivo CAR-T therapies [111,112], protein replacement therapies [113], and stem cell therapies [114] (summarized in Table 2). These advancements position circRNAs at the forefront of a revolutionary wave in nucleic acid-based therapeutics.

### 5.1. CircRNA Vaccines for Viral Infectious Diseases

Viral infections occur when viruses invade host cells, replicate, and trigger inflammatory responses, leading to tissue damage. Current antiviral drugs and preventive vaccines (e.g., for COVID-19 and influenza) face limitations due to the high transmissibility of viruses, immune evasion, and rapid mutation. RNA vaccines have demonstrated remarkable efficacy in preventing and treating viral diseases. As a novel nucleic acid vaccine platform, circRNA vaccines offer unique advantages, including low immunogenicity, high stability, ease of production, and favorable safety profiles. Recent breakthroughs in antigen optimization, delivery system refinement, and immune mechanism exploration have positioned circRNA vaccines as transformative tools for controlling viral diseases.

#### 5.1.1. SARS-CoV-2 and Variants

CircRNA vaccines targeting SARS-CoV-2 and its variants demonstrate exceptional protective efficacy. Qu et al. first developed a circRNA vaccine encoding the trimeric receptor-binding domain (RBD) of the spike protein. This vaccine elicited potent humoral/cellular immunity through highly stable antigen expression, conferring broad-spectrum protection against SARS-CoV-2 and its variants. The study revealed that following LNP delivery, RBD antigen expression levels from circRNA considerably exceeded those from mRNA throughout the expression period. After 28-day storage at 4 °C, LNP-encapsulated circRNA maintained near-complete antigen expression capability, whereas the activity of pseudouridine modification (m1Ψ)-mRNA vaccines decreased by more than half [115]. Seephetdee et al. developed a vaccine prototype based on a circRNA delivery system that successfully expresses the engineered SARS-CoV-2 spike protein VFLIP-X. This antigen incorporates six key mutations (K417N, L452R, T478K, E484K, N501Y, and D614G) into the VFLIP scaffold—which contains five proline substitutions and interchain disulfide bonds to stabilize the trimeric conformation. In murine models, a 5 μg dose of the vaccine induced durable immune responses: Omicron-specific IgG levels remained elevated through week 7 post-boost, demonstrating broad neutralizing activity against multiple variants of concern (VOCs) including Wildtype, Alpha, Beta, Delta, and Omicron while maintaining Th1/Th2 balance. This approach provides a blueprint for addressing rapidly evolving viruses [116].

#### 5.1.2. Monkeypox Virus (MPXV)

Following the 2022 mpox outbreak, Jinge Zhou’s team developed circRNA vaccines encoding four MPXV proteins (A29L, A35R, B6R, and M1R) delivered via LNPs. In mice, all four vaccines elicited humoral and T cell immunity against MPXV. The M1R-encoding vaccine (cirM1R) showed superior humoral responses, while the A35R-encoding vaccine (cirA35R) robustly activated T cell immunity. A tetravalent combination (cirMix4) achieved complete protection against lethal vaccinia virus (VACV) challenge and significantly reduced pulmonary pathology [105].

#### 5.1.3. Zika Virus (ZIKV)

CircRNA vaccines address critical challenges in ZIKV vaccine development, including antibody-dependent enhancement (ADE), low EDIII immunogenicity, and risks associated with dengue co-infection. Xinglong Liu’s team designed a circRNA vaccine encoding ZIKV envelope domain III (EDIII) fused with human IgG1 Fc (EDIII-Fc) and nonstructural protein NS1. The EDIII-Fc circRNA induced robust germinal center reactions, high neutralizing antibodies, and Th1-biased T cell responses in mice, while NS1 circRNA provided additional ADE-independent protection. A single-dose regimen conferred durable protection (≥11 weeks) against lethal ZIKV infection and neurodevelopmental abnormalities without ADE. Optimization of circRNA scaffolds (e.g., HRV-B3 IRES and PABP-binding motifs) further enhanced translational efficiency and immunogenicity [107].

#### 5.1.4. Rabies Virus

Wan et al. developed a lymph node-targeting circRNA vaccine expressing rabies virus glycoprotein G (circRNA-G). This study innovatively created a mannose-modified lipid nanoparticle (mLNP) delivery system, establishing a novel vaccine platform (mLNP-circRNA-G) with dual lymph node-targeting capability and lyophilization stability. Data demonstrated superior immunogenicity of circRNA-induced antibody responses over mRNA in titer, durability, and quality. Following a single 2 μg dose of circRNA-G or mRNA-G in mice, circRNA-G elicited high levels of G-specific IgG geometric mean titers (GMTs), reaching approximately 5.3 × 10^6^ at 5 weeks post-immunization (wpi), with peak IgG levels (at 7 wpi) 396-fold higher than mRNA-G. Similarly, circRNA-G-induced neutralizing antibody (nAb) levels reached 37.7 IU/mL at 4 wpi, exhibiting peak nAb levels (at 20 wpi) 9.2-fold greater than mRNA-G. Notably, mRNA-G not only generated lower IgG and nAb levels than circRNA-G but also, as expected, exhibited significant antibody decline after 4 wpi. Furthermore, mRNA-G maintained low nAb titers from 14 wpi onward, with 20% of mice showing seronegative nAb responses at 18 wpi. In contrast, circRNA-G achieved peak antibody levels at 4 wpi, remained stable through 4–8 wpi, and only began declining after 9 wpi while sustaining higher levels at 20 wpi. Additionally, circRNA-G demonstrated significantly enhanced antibody affinity compared to mRNA-G. Collectively, these data indicate that circRNA-G immunization generates stronger and more durable immune responses than mRNA-G vaccination [118].

#### 5.1.5. Influenza Virus

Yue et al. developed a multivalent circRNA-LNP vaccine encoding neuraminidase (NA) antigens from influenza A (N1, N2) and B (Victoria lineage). This vaccine elicited cross-protective immunity against both matched and mismatched strains (e.g., H1N1, H3N2, influenza B) in mice and activated Th1-biased cellular immunity, offering a novel strategy for the design of a universal influenza vaccine [106].

### 5.2. CircRNA Vaccines for Cancer Therapy

CircRNA vaccines demonstrate transformative potential in cancer immunotherapy, leveraging their unique closed-loop structure to overcome the limitations of conventional mRNA vaccines through enhanced stability and prolonged antigen expression.

#### 5.2.1. High-Efficiency Vaccine Platforms Inducing Antitumor Immune Responses

The stable circular architecture of circRNA prolongs antigen presentation, amplifying antitumor immunity. Li et al. developed a circRNA- LNP vaccine encoding tumor-associated antigens (TAAs), which triggered robust innate and adaptive immune activation in murine models, significantly inhibiting the growth of multiple malignancies. Compared to linear mRNA, circRNA’s resistance to exonuclease degradation enables sustained antigen expression in vivo, promoting dendritic cell (DC) maturation, enhancing antigen presentation, and ultimately driving the expansion of antigen-specific CD8^+^ T cells [130]. Amaya et al. delivered tumor antigen-encoding circRNA via a charge-reversible delivery system, demonstrating DC activation and antigen-specific CD8^+^ T cell proliferation in lymph nodes and peripheral tissues, which led to marked tumor suppression [64]. Similarly, Wang et al. developed an LNP-encapsulated circRNA vaccine that achieved sustained antigen expression in hepatocellular carcinoma models, promoting DC maturation and enhancing T cell cytotoxic activity. The treatment group exhibited >80% tumor suppression. In orthotopic Hepa1-6 liver cancer models, 80% of tumors were completely eradicated in the circRNA-vaccinated cohort, whereas significantly lower eradication rates were observed in PBS (0%) and linear RNA (40%) control groups. Notably, circRNA-vaccinated mice demonstrated prolonged overall survival compared to PBS- or linear RNA-treated groups [108]. Yu et al. (2025) further improved the translation efficiency of circRNA through structural optimization, such as the introduction of IRES within enterovirus A (EV-A) and the optimization of secondary structures. Studies demonstrated that circRNA vaccines induce stronger T cell immune responses than conventional linear mRNA vaccines, exhibiting superior efficacy in tumor prevention and treatment. In B16F10 tumor models, all three vaccine groups elicited elevated IFN-γ+ CD8+ T cell levels, with the CircB16-8 cohort showing the highest mean proportion: CircB16 (1.39%), LinearB16 (0.80%), and m1ΨLinearB16 (1.10%). In TC-1 models, blood samples collected 7 days post-boost vaccination revealed that the CircE6E7 vaccine induced significantly higher proportions of IFN-γ-secreting T cells—8.5-fold greater than unmodified linear RNA (7.52% vs. 0.88%) and 4.5-fold greater than modified linear RNA (7.52% vs. 1.67%). Consistent with antigen-specific T cell induction, CircE6E7 and m1ΨLinearE6E7 vaccines achieved complete tumor eradication, while LinearE6E7 only weakly suppressed tumor growth. Collectively, these results highlight limitations of m1Ψ and demonstrate that circRNA vaccines encoding tumor neoantigens and HPV antigens elicit stronger tumor antigen-specific T cell immunity and superior therapeutic efficacy compared to conventional linear mRNA vaccines in murine models [131]. Additionally, the Li team developed an intranasal circRNA vaccine targeting mucosal immunity. DOTAP-containing LNPs facilitated lung-specific delivery, thereby enhancing the uptake of antigen-presenting cells (APCs). This vaccine relied on type 1 conventional dendritic cells (cDC1s) for antigen presentation, with alveolar macrophages (AMs) synergizing with cDC1s to amplify pulmonary T cell responses. Single-cell RNA sequencing revealed that the vaccine drives remodeling of endogenous T cells, upregulating effector memory T cells (TEM) and cytotoxic markers (GZMA/GZMB) [121].

#### 5.2.2. Targeting the Immunosuppressive Tumor Microenvironment

CircRNA vaccines counteract key immunosuppressive factors. Chen et al. identified the H19-derived immunoregulatory protein (H19-IRP) in glioblastoma (GBM), which recruits myeloid-derived suppressor cells (MDSCs) and tumor-associated macrophages (TAMs) through the activation of CCL2 and Galectin-9, thereby inducing T cell exhaustion. A circRNA vaccine targeting H19-IRP (circH19-vac) elicited cytotoxic T cell responses, significantly suppressing the progression of GBM in murine models [119].

#### 5.2.3. Tumor-Specific circRNAs as Neoantigen Sources

Noncanonical translation of circRNAs provides neoantigens for tumors with low mutational burden. Huang et al. discovered cryptic peptides encoded by circFAM53B in breast cancer and melanoma, presented via HLA-I/II molecules to activate CD4^+^/CD8^+^ T cells. High circFAM53B expression correlated with improved patient survival [117]. Ren et al. identified neoantigenic peptides from circRAPGEF5 and circMYH9 in microsatellite-stable colorectal cancer. ELISpot and cytotoxicity assays confirmed T cell activation and patient-derived organoid elimination. Notably, circMYH9 enrichment in liquid biopsies supports a “detect-to-vaccinate” strategy [120].

#### 5.2.4. Synergistic Combination Therapies

Combining circRNA vaccines with other modalities enhances efficacy. Cai et al. reported that circRNA-loaded DC vaccines combined with gemcitabine increased tumor inhibition rates from 69% (monotherapy) to 89% in pancreatic cancer models by inducing immunogenic cell death (ICD), antigen spreading, and Treg reduction [122]. Similarly, Zhang et al. demonstrated that circRNA vaccines reversed PD-1 blockade resistance in melanoma, prolonging survival. Data revealed that unmodified small circRNAs exhibit superior thermal stability compared to both corresponding linear RNA (liRNA) and 5moU-mRNA-OVA. Single-phase decay modeling estimated half-lives in PBS storage at −20 °C, 4 °C, and 23 °C as 401 days, 78 days, and 16 days for circRNA-SIINFEKL; 143 days, 44 days, and 6 days for 5moU-mRNA-OVA; and merely 2.6 days, <0.5 days, and <0.5 days for liRNA-SIINFEKL, respectively. Consistently, following transfection into DCs, circRNA-Flag sustained Flag expression for ≥7 days, whereas 5moU-mRNA-fLuc protein expression lasted <3 days. In C57BL/6 mice receiving escalating vaccine doses, all formulations elicited dose-dependent SIINFEKL^+^CD8^+^ T cell responses, with circRNA-SIINFEKL inducing the highest proportion of SIINFEKL^+^CD8^+^ T cells among peripheral blood mononuclear cell (PBMC) CD8^+^ T cells. Under identical conditions (2 × 1-μg), circRNA-SIINFEKL generated 2.6-fold and 7.3-fold higher proportions of SIINFEKL^+^CD8^+^ T cells in PBMCs than 5moU-mRNA-SIINFEKL and ψ-mRNA-SIINFEKL, respectively. Regarding safety, circRNA-SIINFEKL caused less body weight loss and faster recovery compared to benchmark mRNA vaccines. Small circRNA-SIINFEKL also demonstrated reduced PKR phosphorylation (pPKR) levels in cells relative to 5moU-mRNA-OVA [59]. In the study by Li et al., circRNA vaccines, ACT, and combination therapy were administered separately. Results revealed superior tumor control efficacy in the combination group, where the vaccine not only expanded endogenous T cells but also enhanced the effector function of adoptively transferred T cells. Moreover, studies demonstrated that circRNA maintains significantly higher protein expression levels than linear RNA following storage at both 4 °C and 37 °C [121].

### 5.3. CircRNA Therapeutics for Autoimmune and Metabolic Disorders

While current research on circRNA therapies predominantly focuses on vaccines for viral infections and cancers, their potential in autoimmune and metabolic diseases is increasingly recognized [132,133].

As reported, circRNAs exhibit high stability and low immunogenicity due to their closed circular structure, enabling unique conformational regulation of key immune factors. circRNAs inhibit aberrant PKR activation through 16–26 bp ds regions that bind dsRNA-activated PKR. In systemic lupus erythematosus (SLE) patients, global reduction in circRNAs correlates with excessive PKR phosphorylation, while supplementation of ds-structure-containing circRNAs reverses abnormal PKR signaling in peripheral blood mononuclear cells, demonstrating intrinsic immunomodulatory functions [48]. Subsequent studies further validated circRNA efficacy. Engineered ds-cRNA synthesized via optimized self-splicing intron (PIE) technology significantly suppressed PKR activity in a psoriatic mouse model. This ds-cRNA (EPIC), delivered via lipid nanoparticles to the spleen, reduced splenocyte PKR phosphorylation and attenuated epidermal thickening and inflammatory cytokines (e.g., IL-17, IL-23), without eliciting notable immune responses. Single-molecule imaging revealed that EPIC stably binds PKR to prevent dimerization activation, outperforming linear RNA inhibitors. circRNA-overexpressing mouse models confirmed their capacity to alleviate early inflammatory responses, laying groundwork for clinical translation [123]. Notably, on June 11, 2025, Capstan Therapeutics announced the commencement of a Phase 1 clinical trial for its CAR-T cell therapy CPTX2309 (anti-CD19 in vivo CAR-T) in patients with B cell-mediated autoimmune diseases. The preclinical results were published, demonstrating the use of targeted lipid nanoparticles (tLNPs) to deliver mRNA in vivo for direct reprogramming of T cells to generate CAR-T cells for the treatment of B cell malignancies and autoimmune diseases [134]. This research opens new avenues for applying circRNA therapeutics to autoimmune disorders.

Zhao et al. demonstrated that mitochondria-localized circRNA SCAR directly binds ATP synthase β subunit (ATP5B) in non-alcoholic steatohepatitis (NASH), disrupting cyclophilin D (CypD)-mPTP interaction to inhibit pathological mPTP opening, thereby reducing mitochondrial ROS (mROS) release and ultimately alleviating fibroblast activation and hepatic fibrosis. Lipid overload suppresses PGC-1α-mediated circRNA SCAR transcription via ER stress-CHOP signaling. Mitochondria-targeted nanoparticles (mito-NPs), engineered with pH-responsive polymers and TPP-modified cationic peptides, achieved >90% mitochondrial delivery efficiency. In vivo, mito-NP-delivered circRNA SCAR significantly improved high-fat diet-induced insulin resistance and liver fibrosis without toxicity. Clinical analyses confirmed an inverse correlation between circRNA SCAR levels and NASH fibrosis severity [35]. In diabetic wound healing, ADSC-derived exosomes deliver circ-Snhg11 to mitigate hyperglycemia-induced endothelial damage and promote M2 macrophage polarization [124]. Hu et al. further encapsulated hypoxia-preconditioned ADSC exosomes in methacryloyl gelatin (GelMA) hydrogels, demonstrating that circ-Snhg11 enhances endothelial function via miR-144-3p/NFE2L2 pathway inhibition to accelerate diabetic wound healing [126]. Tang et al. revealed that BMSC exosome-delivered circ-Snhg11 activates SLC7A11/GPX4 signaling to inhibit endothelial progenitor cell ferroptosis, thereby improving diabetic neovascularization [125]. Liu et al. innovatively encapsulated VEGF-A circRNA in U-105 nanoparticles, achieving sustained two-week protein expression from a single dose with superior healing efficacy over linear mRNA and recombinant protein therapies [113]. Shi et al. identified circ-IGF1R-mediated regulation of HK2/VEGFA axis through competitive miR-503-5p binding to enhance hypoxia-preconditioned exosome efficacy [127]. Additionally, circCDK13-loaded small extracellular vesicles (sEVs) activated m6A-dependent CD44/c-MYC signaling to achieve skin appendage regeneration in diabetic rat models [128].

### 5.4. Advances in Other Protein Replacement Therapies via circRNA-Encoded Functional Proteins

The discovery of circRNAs encoding functional proteins has opened new avenues for protein replacement therapies [21]. For instance, hsa_circ_0002301 has been demonstrated to encode a novel small protein, HECTD1-463aa, which competitively binds to HECTD1 and suppresses its ubiquitination-mediated degradation of GPX4, thereby regulating ferroptosis in gastric cancer cells [135]. In intrahepatic cholangiocarcinoma (ICC) models, the translation product of circPCSK6, circPCSK6-167aa, competitively binds to the RING domain of the E3 ubiquitin ligase RBBP6, inhibiting K48 ubiquitination of IKBα and preventing its degradation. This mechanism suppresses NF-κB nuclear translocation and downstream oncogenic activation [136]. Peng et al. demonstrated that circCOPA encodes the COPA-99aa protein to suppress malignant progression in GBM and enhance chemosensitivity to temozolomide (TMZ). Overexpression of circCOPA inhibits GBM cell proliferation, migration, and invasion in vitro, while in vivo studies confirm tumor growth suppression. Mechanistically, COPA-99aa disrupts the heterodimer formation of NONO and SFPQ, impairing DNA double-strand break repair and amplifying TMZ-induced DNA damage [137].

Wang et al. injected MDA-MB-231 cells stably transfected with circSEMA4B or its mutants into the mammary fat pads of nude mice, revealing that circSEMA4B encodes a novel protein, SEMA4B-211aa, which is downregulated in breast cancer. Overexpression of SEMA4B-211aa significantly suppresses tumor cell proliferation and migration [129]. In diabetic foot ulcer (DFU) treatment, Liu et al. developed an ionizable lipid U-105-based nanoparticle (U-LNP) delivery system to encapsulate VEGF-A-encoding circRNA. A single topical administration of U-LNP/circRNA enabled sustained VEGF-A protein expression at the wound site for over one week, markedly promoting angiogenesis and epidermal regeneration. Complete wound healing in diabetic mice was achieved within 12 days, outperforming linear mRNA, recombinant protein, and traditional ALC-0315 lipid-based delivery systems [113]. Similar strategies show promise in oncology. For lung cancer, the H1L1A1B3 lipid nanoparticle, optimized via high-throughput screening, achieved efficient IL-12 circRNA delivery. A single intratumoral injection induced significant CD8+ T cell infiltration and robust antitumor immunity, leading to marked tumor regression in Lewis lung carcinoma models. This platform quadrupled circRNA transfection efficiency compared to standard ALC-0315 lipids while minimizing systemic toxicity through tailored lipid composition [110].

## 6. Clinical Trials of circRNA-Based Therapeutics

The unique covalently closed circular structure of circRNAs confers superior biological stability compared to linear mRNA, positioning them as a promising therapeutic modality. Recent years have witnessed substantial progress in translating circRNA from fundamental research to clinical applications, with multiple candidate therapies entering clinical trials targeting oncology vaccines, protein replacement therapeutics, and rare diseases. This section summarizes circRNA-related clinical investigations (Table 3).

In 2023, the Second Affiliated Hospital of Bengbu Medical College initiated the first-in-human trial (NCT05784137) of a circRNA COVID-19 vaccine (TI-0010; Therorna lnc., Beijing, China), evaluating safety and immunogenicity in adults (18–59 years). This randomized, double-blind, placebo-controlled study assessed an LNP-formulated circRNA encoding the SARS-CoV-2 spike RBD domain, generating critical translational data for circRNA vaccination strategies.

In 2024, Sun Yat-sen Memorial Hospital advanced a clinical trial leveraging the noncanonical translation function of circRNA. The study utilized cryptic peptides (219aa) encoded by CircFAM53B, a circRNA highly expressed in breast cancer, as neoantigens. This trial evaluated the safety, tolerability, and efficacy of a CircFAM53B-219aa-loaded DC vaccine, both as monotherapy and in combination with the PD-1 inhibitor camrelizumab, in HER2-negative advanced breast cancer patients. Targeting patients with limited survival benefits from second-line therapies, preclinical studies demonstrated that CircFAM53B-219aa-loaded DCs activated antigen-specific T cell responses and significantly suppressed tumor growth in CircFAM53B-high animal models.

In 2025, HM2002—a circRNA therapy independently developed by Shanghai CirCode Biotherapeutics—received clinical trial approval (IND) from China’s NMPA for IHD treatment, marking China’s first NMPA-approved circRNA biologic. Back in August 2024, an investigator-initiated trial (IIT) led by Prof. Zhao Qiang’s team at Ruijin Hospital, Shanghai Jiao Tong University administered 5 mg HM2002 via intraoperative epicardial myocardial injection to ischemic heart disease (IHD) patients undergoing coronary artery bypass grafting (CABG). By October 2024, all patients completed dosing, with preliminary results indicating favorable recovery, no drug-related adverse events (e.g., severe allergies, myocardial injury), and significant post-operative improvements in cardiac function parameters (e.g., left ventricular ejection fraction).

In 2025, RiboX Therapeutics’ RXRG001 became the first circRNA therapy approved by the U.S. FDA for clinical trials and the only circRNA drug approved for radiation-induced xerostomia. This first-in-human Phase I/IIa dose-escalation study evaluates the safety, tolerability, and preliminary efficacy of RXRG001 via parotid duct injection for xerostomia management. The trial comprises two parts: Part 1 (open-label, single-arm) tests single ascending doses (three cohorts) and multiple ascending doses (three cohorts) in unilateral parotid injections; Part 2 (randomized, double-blind, placebo-controlled) compares multiple ascending doses (three cohorts) in bilateral parotid injections.

## 7. Conclusions and Perspective

CircRNA therapeutics represent a paradigm shift in nucleic acid medicine, uniquely combining the intrinsic stability of RNA with the potent regulatory and functional capabilities of proteins. The advent of sophisticated self-splicing ribozyme systems (e.g., optimized PIE) and AI-guided design platforms (e.g., circDesign) has significantly enhanced circRNA synthesis efficiency, translational capacity, and stability. Concurrently, modular delivery platforms, particularly LNPs, have accelerated the clinical translation of this promising modality. Current research robustly demonstrates that circRNA vaccines effectively orchestrate both humoral and cellular immunity, showcasing significant potential not only in combating viral infections and presenting tumor neoantigens but also in modulating autoimmune responses. Preliminary clinical trials, bolstered by advanced molecular insights from cryo-EM and single-cell sequencing, are establishing an encouraging initial safety profile for these novel therapeutics.

To fully realize the clinical potential of circRNA and transition from promising platform to established therapeutic paradigm, several critical knowledge gaps and challenges demand focused investigation:

**Scalable Manufacturing and Quality Control:** Achieving cost-effective, large-scale production requires standardized protocols specifically tailored for circRNA. Key challenges include maximizing circularization yield with minimal immunogenic impurities (e.g., residual linear RNA, dsRNA contaminants) and establishing robust, scalable purification processes. Developing continuous-flow bioreactor systems integrating in-line circularization monitoring (e.g., via capillary electrophoresis or HPLC-MS) and implementing affinity-tag-based purification strategies (e.g., using optimized aptamers) could streamline production and ensure batch-to-batch consistency. Harmonizing quality control standards with evolving regulatory guidelines (building upon existing mRNA frameworks from CDE.NMPA, WHO, USP) is paramount.

**Precision Delivery and Pharmacokinetics:** While systemic delivery via LNPs shows promise, achieving tissue- and cell-type specificity remains a hurdle. Furthermore, a deeper understanding of circRNA pharmacokinetics, biodistribution, and long-term persistence in target versus off-target tissues is essential, especially for chronic applications. Engineering “smart” LNPs incorporating tissue-homing peptides (e.g., cardiac-targeting, brain-penetrating) or microenvironment-responsive elements (e.g., pH-sensitive lipids in tumors, enzyme-cleavable linkers in inflamed tissues) warrants exploration. Utilizing advanced in vivo imaging techniques (e.g., bioorthogonal labeling of circRNA combined with PET/CT or SPECT imaging) in relevant disease models (e.g., cancer xenografts, autoimmune models like SLE or psoriasis in mice/NHP) is crucial to map biodistribution and correlate it with efficacy/toxicity.

**Immune Modulation: Balancing Efficacy and Tolerance:** While low intrinsic immunogenicity is an advantage, harnessing or suppressing circRNA-triggered immune responses contextually is vital. The dual role of circRNA—potentially acting as an adjuvant in vaccines but needing stealth in protein replacement or chronic therapy—requires precise control. The mechanisms underlying circRNA’s interactions with immune sensors (e.g., RIG-I, PKR, TLRs) and its impact on adaptive immunity need further quantification. Developing quantitative predictive models of circRNA immunogenicity, integrating sequence features (motifs, modifications like m6A), structural elements, and delivery context using machine learning trained on in vitro (immune cell assays) and in vivo immunogenicity data. Exploring strategies like targeted suppression of specific degradation pathways implicated in unwanted immune activation (e.g., DIS3 inhibition to enhance stability without broad RNase suppression, based on recent findings [53]) could fine-tune immunogenicity. Rigorously evaluating the long-term immunological consequences, including potential tolerance induction or autoimmunity risk, in chronic dosing studies (e.g., in NHP models for protein replacement therapies) is essential.

**Long-Term Safety and Endogenous RNA Crosstalk:** Comprehensive assessment of the long-term safety profile of exogenous circRNA, particularly regarding integration risks (minimal but requires confirmation) and potential interference with endogenous RNA metabolism and function, is still nascent. Establishing dedicated long-term (e.g., 6–12 month) toxicology studies in relevant animal models, incorporating detailed histopathology and molecular analysis (e.g., transcriptome sequencing to assess off-target effects on endogenous circRNA/miRNA networks). Elucidating the interactions between therapeutic circRNA and cellular RNA decay machinery (e.g., DIS3, exosome complex) and quality control pathways will inform safer design.

**Synergistic Therapeutic Strategies:** The potential of circRNA extends beyond standalone vaccines or protein expression. Exploring innovative combinations, such as circRNA encoding inducible factors paired with CRISPR-based gene editing tools for in situ tissue regeneration, or circRNA vaccines co-delivered with immune checkpoint modulators or oncolytic viruses to overcome immunosuppressive tumor microenvironments, represents a frontier. Optimizing circRNA for sustained, localized expression of therapeutic proteins (e.g., leveraging findings on VEGF-A circRNA wound healing [113]) within engineered biomaterials also holds significant promise.

The convergence of synthetic biology, advanced delivery technologies, and artificial intelligence is rapidly transforming circRNA platforms. AI will be instrumental not only in optimizing sequences and predicting immunogenicity but also in deciphering complex circRNA–protein interaction networks and enabling the design of programmable therapeutic circuits. Addressing the outlined challenges through focused and collaborative research—establishing standardized frameworks for design, production, and evaluation—will be critical over the next five years. Successfully navigating these hurdles will catalyze the transition of circRNA therapeutics from compelling proof-of-concept studies to impactful clinical applications, ultimately solidifying its position as a versatile and transformative therapeutic paradigm for a wide spectrum of human diseases.

## Figures and Tables

**Figure 1 vaccines-13-00821-f001:**
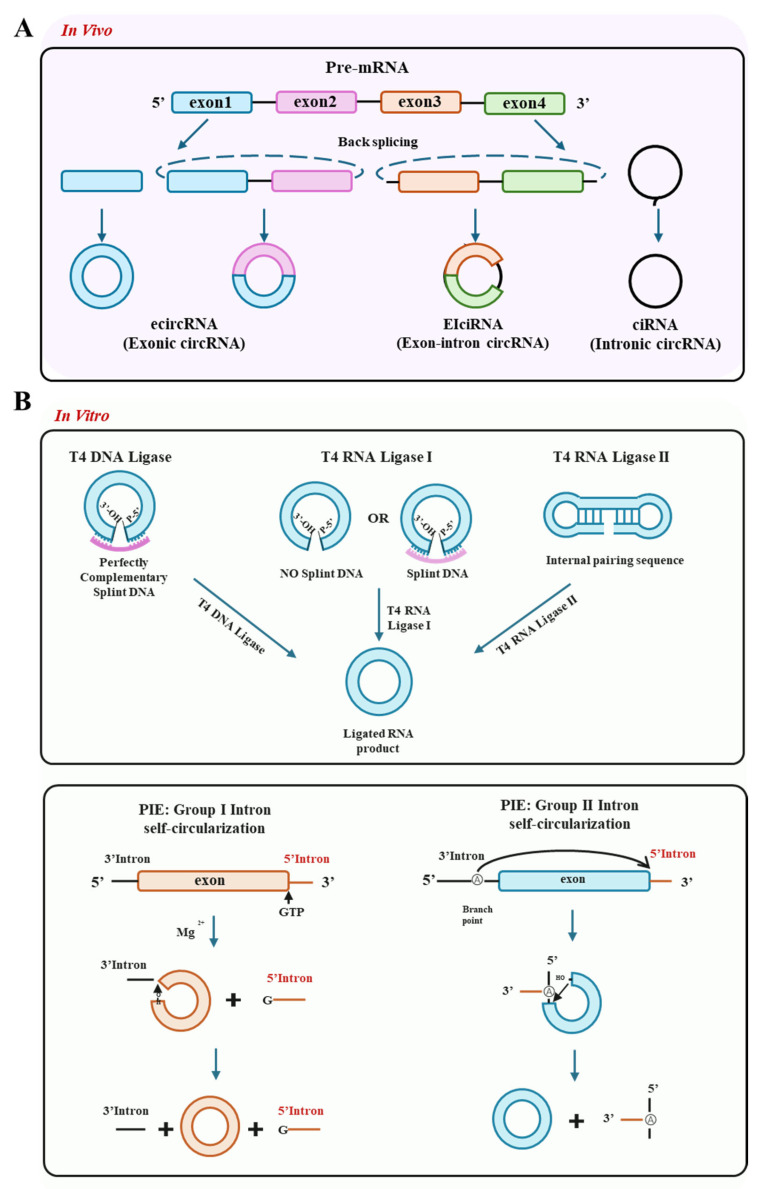
The synthesis of circRNA. (**A**) Endogenous circRNAs are primarily generated through back-splicing of pre-mRNA. (**B**) In vitro circRNA synthesis using T4 ligase and PIE strategies (adapted with permission from Wei et al., 2025 [23]).

**Figure 2 vaccines-13-00821-f002:**
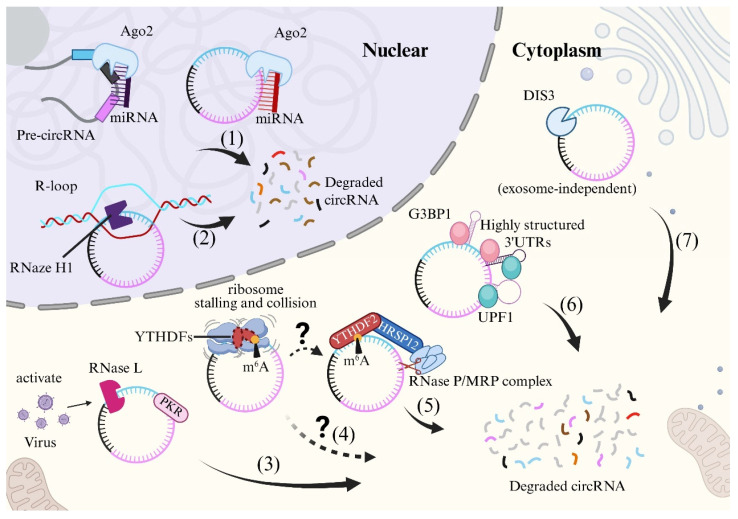
Possible degradation pathway of circRNA. (1) Ago2 participates in circRNA degradation. (2) RNase H1-mediated circRNA degradation. (3) RNase L-mediated circRNA degradation. (4) m^6^A-modified RNAs could induce ribosome stalling and collision which recruits YTHDFs to initiate degradation. (5) m^6^A-modified circRNAs are recognized by YTHDF2 and degraded via HRSP12-RNase P/MRP complexes. (6) Structure-mediated circRNA degradation. (7) DIS3-dependent circRNA degradation pathway. Solid arrows denote experimentally validated degradation pathways, whereas dashed question-marked arrows indicate putative degradation mechanisms.

**Figure 3 vaccines-13-00821-f003:**
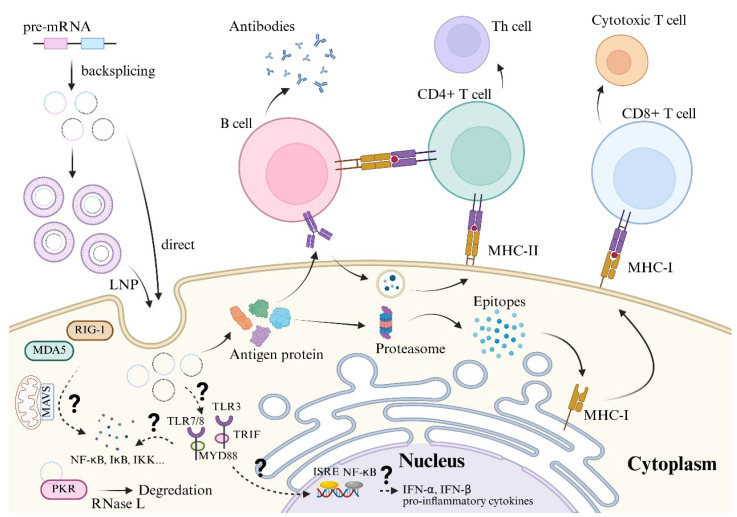
Immune responses to circRNA. Exogenous circRNA delivered into the body by LNP or direct injection can trigger a series of immune responses, including innate immune responses and adaptive immune responses. When exogenous circRNA has very high purity, it does not induce innate immune responses. Solid black arrows indicate immune responses triggered by exogenous circRNA, while dashed arrows indicate potential innate immune responses or innate immune responses triggered by impure circRNA entering the body.

**Figure 4 vaccines-13-00821-f004:**
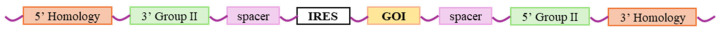
Key components of the PIE-based strategy.

**Figure 5 vaccines-13-00821-f005:**
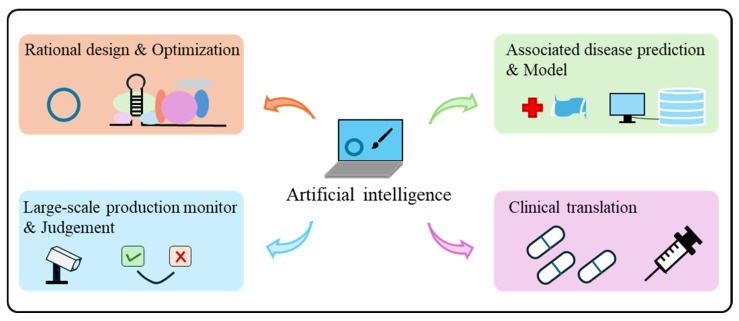
AI drives the development of circRNA vaccines/medicines. From circRNA medicine design, disease prediction, and model construction to large-scale industrial production and clinical application, AI will play a very important role in every stage of the process.

**Table 1 vaccines-13-00821-t001:** Advantages and disadvantages of various vaccines.

Vaccine Type	Profile	Advantages	Disadvantages
Inactivated	Pathogen inactivated in entirety	Mature, traditional technology High safety profile (non-viable pathogen) Broad antigenic spectrum (reduces impact of variants)	Requires adjuvants to enhance immunogenicity Requires 2–3 booster doses Complex production (requires high-containment labs)
Live-attenuated	Live pathogen with attenuated virulence	Potent and long-lasting immune response Often confers lifelong immunity with 1–2 doses	Risk of virulence reversion (causing disease) Contraindicated for immunocompromised individuals High-security facilities required, elevated costs
Subunit	Specific pathogen components (protein/polysaccharide/VLP)	High safety profile (no intact pathogen) Strong stability Amenable to large-scale production	Requires adjuvants and booster doses Susceptible to antigenic variation (may compromise efficacy) Complex development (requires precise antigen selection)
Viral Vector	Antigen gene delivered via harmless viral vector	Mimics natural infection Potent immunity (often single-dose effective) Rapid development cycle	Complex manufacturing process Vector immunity can attenuate efficacy Rare but serious side effects (e.g., thrombosis)
DNA	Antigen gene inserted into plasmid DNA vector	High stability (storage at ambient temperatures) No live virus required (enhanced safety) Low production cost	Requires nuclear entry for transcription Potential risk of genomic integration Low immunogenicity (suboptimal potency)
mRNA	Antigen-encoding mRNA encapsulated in lipid nanoparticles	No risk of genomic integration Rapid adaptation to variants Relatively low production cost	Instability and susceptibility to degradation Stringent cold-chain storage requirements Potential inflammatory risks
circRNA	Synthetic circRNA encapsulated in lipid nanoparticles	No risk of genomic integration High intrinsic stability Safety (low inherent immunogenicity) Self-adjuvant effect Prolonged antigen expression	Immature manufacturing processes Low circularization efficiency Limited clinical data

**Table 2 vaccines-13-00821-t002:** Preclinical applications of circRNA medicines.

CircRNA	Expressed Protein	Disease/Pathogen	Application Field	Mechanism/Function	Reference
circRNA RBD-Delta	Spike trimeric RBD (Delta/Omicron)	COVID-19	Vaccine prevention	Induces neutralizing antibodies and Th1-skewed cellular immunity, providing broad-spectrum protection against variants	[115]
circRNA VFLIP-X	VFLIP-X spike protein (K417N mutations)	SARS-CoV-2 variants	Vaccine prevention	Elicits broad neutralizing antibodies and balanced Th1/Th2 responses against multiple VOCs/VOIs	[116]
circFAM53B	Cryptic peptides (ALFRLTNRA/RTAHYGTGR)	Breast cancer, melanoma	Universal cancer vaccine	Encodes tumor-specific antigens (TSAs) via HLA-I/II dual presentation, activating CD8+/CD4+ T cells	[117]
circRNA EDIII-Fc + NS1	EDIII-Fc and NS1 proteins	Zika virus	Vaccine prevention	Induces neutralizing antibodies and germinal center (GC) reactions, avoiding DENV ADE effect with single-dose efficacy	[107]
circRNA-G	Rabies virus glycoprotein G (RABV-G)	Rabies	Vaccine prevention	Enhances humoral immunity (high IgG/neutralizing antibodies) and lymph node-targeted delivery for prolonged antigen expression	[118]
circRNA-NA (N1/N2/IBV)	Neuraminidase (N1, N2, influenza B)	Influenza	Vaccine prevention	Induces broad neutralizing antibodies and Th1-skewed immunity against H1N1/H3N2/Victoria/Yamagata strains	[106]
cirA29L/cirA35R/cirB6R/cirM1R	A29L, A35R, B6R, M1R proteins	Monkeypox	Vaccine prevention	Elicits neutralizing antibodies and T cell responses via multivalent antigens, reducing tissue viral load	[105]
H19-IRP	H19-IRP (protein encoded by lncRNA)	Glioblastoma	Cancer immunotherapy	Activates CCL2/Galectin-9 transcription to recruit MDSCs/TAMs; triggers T cell responses as a TAA	[119]
circRNA-PTPN2	PTPN2 (neoantigen)	Hepatocellular carcinoma	Neoantigen vaccine	Activates DC maturation and T cell responses via circRNA-LNP delivery, enhancing tumor cell targeting	[108]
circRAPGEF5, circMYH9	Tumor-specific cryptic peptides	Colorectal cancer	Liquid biopsy-driven therapy	Presented via HLA-A*11:01, inducing T cell-mediated tumor organoid clearance	[120]
circRNA-LNP	SIIINFEKL (OVA antigen)	Lung cancer	Mucosal immunotherapy	Enhances antigen-specific T cell responses via cDC1s and alveolar macrophages, reducing systemic toxicity	[121]
circRNA-FS (FAPα/survivin)	FAPα, survivin	Pancreatic cancer	Chemo-immunotherapy combination	Enhances DC vaccine antigen expression, induces ICD, synergizes with gemcitabine to suppress Tregs	[122]
Small circRNA (<300 nt) (e.g., circRNA-SIINFEKL)	Peptide antigens (e.g., SIINFEKL)	Low-immunogenic tumors (e.g., melanoma)	Long-term immune memory induction	High stability (half-life > 7 days), low PKR activation, synergizes with immune checkpoint inhibitors	[59]
circRNA SCAR	-	Non-alcoholic steatohepatitis	Metabolic disease/Liver disease	Binds ATP5B and inhibits mitochondrial permeability transition pore (mPTP) opening, reduces mitochondrial ROS output, alleviates fibroblast activation and inflammation	[35]
circPOLR2A (EPIC)	-	Psoriasis	Immune modulation/inflammation control	Stabilizes PKR binding to inhibit its activity, attenuates IFN-α signaling and dsRNA-mediated inflammatory responses	[123]
circ-Snhg11	-	Diabetes mellitus	Wound healing	Inhibits hyperglycemia-induced endothelial damage via miR-144-3p/HIF-1α axis, induces M2-like macrophage polarization	[124]
circ-Snhg11	-	Diabetes mellitus	Diabetic wound healing	Enhances SLC7A11/GPX4-mediated anti-ferroptosis signaling through miR-144-3p sponge effect, promotes angiogenesis	[125]
circ-Snhg11	-	Diabetes mellitus	Angiogenesis	Activates miR-144-3p/NFE2L2/HIF1α pathway to suppress oxidative stress and improve endothelial function with vascular regeneration	[126]
circ-IGF1R	-	Diabetes mellitus	Diabetic foot ulcer	Upregulates HK2 and VEGFA expression via miR-503-5p sponge adsorption, enhances angiogenesis while reducing apoptosis	[127]
circCDK13	-	Diabetes mellitus	Wound healing/regenerative medicine	Interacts with IGF2BP3 in m6A-dependent manner to enhance CD44 and c-MYC expression, promoting cutaneous cell proliferation/migration	[128]
VEGF-A circRNA	VEGF-A	Diabetes mellitus	Diabetic foot ulcer	Achieves sustained VEGF-A protein expression via lipid nanoparticle delivery to promote angiogenesis	[113]
IL-12 circRNA	IL-12	Lung cancer	Immunotherapy	Delivers circRNA-encoded IL-12 through H1L1A1B3 LNPs, activates immune response, increases CD8+ T cell infiltration, inhibits tumor growth	[113]
circSEMA4B	SEMA4B-211aa	Breast cancer	Cancer therapy	Encodes SEMA4B-211aa to suppress PI3K/AKT pathway via miR-330-3p/PDCD4 axis-mediated inhibition of AKT phosphorylation	[129]

**Table 3 vaccines-13-00821-t003:** Clinical trials of circRNA therapies.

Study Title	Year	Location	Sponsor	Study Status	Study ID	Data From
A Study to Evealuate Safety and Immunogenicity of TI-0010 SARS-CoV-2 Vaccine in Healthy Adults	2023	China	National Drug Clinical Trial Institute of the Second Affiliated Hospital of Bengbu Medical College	RECRUITING	NCT06205524	clinicaltrials.gov
A Single Arm Clinical Study of Dendritic Cell Vaccine Loaded With CircRNA Encoding Cryptic Peptide for Patients With HER2-negative Advanced Breast Cancer	2024	China	Sun Yat-Sen Memorial Hospital of Sun Yat-Sen University	NOT_YET_RECRUITING	NCT06530082	clinicaltrials.gov
First-in-Human Pilot Study of Epicardial CircRNA-HM2002 Injection in CABG for Ischemic Heart Failure	2024	China	Ruijin Hospital	ACTIVE_NOT_RECRUITING	NCT06621576	clinicaltrials.gov
HM2002 Injection	2025	China	Shanghai CirCode Biomed Co., Ltd, Shanghai, China.	UNKONW	CXSL2400740	Center for Drug Evaluation of NMPA
Study of CircRNA Treatment in Patients with Radiation Induced Xerostomia-1 (RXRG001)	2025	China	RiboX Therapeutics Ltd, Shanghai, China.	RECRUITING	NCT06714253	clinicaltrials.gov

## Data Availability

No new data were created or analyzed in this study.

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
