# Peer review of "Harnessing the Loop: The Perspective of Circular RNA in Modern Therapeutics"

_vaccines, 2025, doi:10.3390/vaccines13080821_

Round 1

Reviewer 1 Report

Comments and Suggestions for Authors

The manuscript by Zhao et al. describes the uses and potential uses of circular RNA in therapeutics. They describe key aspects of synthesis, stability, production, and delivery platforms.

Overall, it is a comprehensive review that discusses recent advances in this novel field. The manuscript is written in a clear and easy-to-follow manner, making it straightforward to understand. References are complete and updated. Tables and figures are clear and complete. The available literature supports conclusions. 

However, it would be beneficial if the authors could include a table comparing the advantages and disadvantages of their method with traditional vaccination methods, such as mRNA vaccination, DNA vaccination, recombinant protein vaccination, and attenuated pathogens. 

I have no further issues to discuss regarding this manuscript, and I believe it should be published in the Journal after minor revisions.

Author Response

Comments:However, it would be beneficial if the authors could include a table comparing the advantages and disadvantages of their method with traditional vaccination methods, such as mRNA vaccination, DNA vaccination, recombinant protein vaccination, and attenuated pathogens.
Response:Table 1 has been listed in section 5 as suggested. It compares advantages and disadvantages of circRNA vaccine with other vaccination methods.

Reviewer 2 Report

Comments and Suggestions for Authors

The review, which covers the principles and uses of circRNA technology, is current and thoroughly studied. All things considered, this overview of the circRNA sector is strong and relevant, and it will be a great addition to vaccines. A substantial and rapidly evolving corpus of literature has been expertly synthesized by the writers.

  1. The abstract is great, but it's a bit combobulated. I'd suggest you to break the single long paragraph into two compact ones for an easier reading.
  2. It would be beneficial to go beyond enumerating the pathways and talk about potential interactions in Section 2.2 on circRNA degradation. Do they compete with one another? Particular to a cell type? It would be quite beneficial to put a bit more synthesis here.
  3. The significance of purification in immunogenicity is a crucial topic that has to be underlined more (Section 2.3). It would clear up a lot of misunderstandings for readers, especially those who are unfamiliar with the topic, to state explicitly that pollutants are mostly to blame for the contradictory claims found in the literature.
  4. For the AI section, I recommend restructuring it with subheadings. Grouping the applications (e.g., Sequence Design, Manufacturing, etc.) would give it a clearer narrative flow.
  5. When focusing on the PIE synthesis method (Section 3), it would be good to add a sentence justifying why it was chosen over others (like T4 ligase) and briefly mention its limitations for a more balanced perspective.
  6. A minor point: for the citations from 2025, it would be helpful to clarify if they are pre-prints or "in press" publications. This just gives the reader proper context on their peer-review status.
  7. The text contains typographical errors and inconsistent terms (e.g., "Vactines" instead of "Vaccines" in several page headings, "breast mRNA constructs" on Page 2 with the presumed intention to say "linear mRNA constructs"). Check carefully and standardise terms (e.g., have "circRNA" used consistently instead of alternating between "circular RNA").
  8. The conclusion (Page 21) is unnecessary and doesn't contain concrete, actionable future research directions. Make an enumeration of gaps in current knowledge, e.g., challenges to targeted delivery or prolonged safety, and propose concrete experimental approaches (e.g., in vivo models or novel delivery systems) to address them.
  9. The review must cover a broad range of applications, but maybe emphasize new information or unique contributions of circRNA therapeutics compared to earlier reviews. For instance, describe how circRNA vaccines address some limitations of mRNA vaccines (e.g., cold-chain) with example instances or evidence.
  10. The English language needs more improvement.

Author Response

Comments 1:The abstract is great, but it's a bit combobulated. I'd suggest you to break the single long paragraph into two compact ones for an easier reading.

Response 1:The abstract has been revised to achieve a more coherent flow. However, following the journal's formatting guidelines, having the abstract split into two paragraphs appears not ideal. Accordingly, we have used 'In this review' to structure the content into two parts as suggested: the preceding part before this phrase introduces the background, while the subsequent part outlines the focus of this review, highlighting its key strengths and significant contributions.

Comments 2:It would be beneficial to go beyond enumerating the pathways and talk about potential interactions in Section 2.2 on circRNA degradation. Do they compete with one another? Particular to a cell type? It would be quite beneficial to put a bit more synthesis here.

Response 2:We have added a brief discussion of potential interactions between circRNA degradation pathways in section 2.2, which has been highlighted in the revised manuscript.

Comments 3:The significance of purification in immunogenicity is a crucial topic that has to be underlined more (Section 2.3). It would clear up a lot of misunderstandings for readers, especially those who are unfamiliar with the topic, to state explicitly that pollutants are mostly to blame for the contradictory claims found in the literature.

Response 3:We briefly outline the contaminants in the purification process of circRNA in section 2.3 and explained that the contamination during the purification procedure is the main attribution for the contradictory claims. All modifications made in this section are highlighted in the revised manuscript.

Comments 4:For the AI section, I recommend restructuring it with subheadings. Grouping the applications (e.g., Sequence Design, Manufacturing, etc.) would give it a clearer narrative flow.

Response 4: In the AI section, we have added subtitles: The rational design, Model construction and databases, Manufacturing and Clinical translation. These have been highlighted in the text.

Comments 5:When focusing on the PIE synthesis method (Section 3), it would be good to add a sentence justifying why it was chosen over others (like T4 ligase) and briefly mention its limitations for a more balanced perspective.

Response 5: We have added the reasons for choosing the PIE strategy in the first paragraph of Section 3, and briefly mentioned its limitations. This sentence is highlighted in the text.

Comments 6:A minor point: for the citations from 2025, it would be helpful to clarify if they are pre-prints or "in press" publications. This just gives the reader proper context on their peer-review status.

Response 6: We have added the status “pre-print” to all citations on the bioRxiv platform and highlighted them in the references. In addition, the status of the remaining literature is currently “published.”

Comments 7:The text contains typographical errors and inconsistent terms (e.g., "Vactines" instead of "Vaccines" in several page headings, "breast mRNA constructs" on Page 2 with the presumed intention to say "linear mRNA constructs"). Check carefully and standardise terms (e.g., have "circRNA" used consistently instead of alternating between "circular RNA").

Response 7: (1) Regrading "Vactines" spelling: We confirm that a comprehensive check for the term "vaccine" was performed. No instances of the misspelling "vactine" were found anywhere in the manuscript. (2) Regarding "breast mRNA constructs" phrasing (Page 2): The manuscript uses the term "linear mRNA construction"; a rigorous search confirmed that "breast mRNA construction" does not appear anywhere in the text. (3) Regarding "circular RNA" formatting: We have thoroughly reviewed the manuscript text. As per standard nomenclature, the full term "circular RNA" is retained for its first occurrence in the Title, Abstract, and main Body text. All subsequent instances have been changed to "circRNA". These modifications have been clearly highlighted within the manuscript. In summary, we have meticulously checked the entire manuscript text. All necessary corrections have been implemented and are highlighted in the manuscript.

Comments 8:The conclusion (Page 21) is unnecessary and doesn't contain concrete, actionable future research directions. Make an enumeration of gaps in current knowledge, e.g., challenges to targeted delivery or prolonged safety, and propose concrete experimental approaches (e.g., in vivo models or novel delivery systems) to address them.

Response 8: The 'conclusion and perspective' section has been revised in accordance with the reviewer's suggestion and highlighted in the text. In this section, we have completely restructured the conclusion to replace general statements with enumerted, actionable future priorities, explicitly addressing: (1) Scalable Manufacturing and Quality Control; (2) Precision Delivery and Pharmacokinetics; (3) Immune Modulation: Balancing Efficacy and Tolerance; (4) Long-Term Safety and Endogenous RNA Crosstalk; (5) Synergistic Therapeutic Strategies.

Comments 9:The review must cover a broad range of applications, but maybe emphasize new information or unique contributions of circRNA therapeutics compared to earlier reviews. For instance, describe how circRNA vaccines address some limitations of mRNA vaccines (e.g., cold-chain) with example instances or evidence.

Response 9: We have revised the text to: (1) explicitly contrast circRNA with mRNA vaccines, particularly their breakthrough in cold-chain reduction ; (2) link thermostability to functional benefits: sustained antigen expression enables durable immunogenicity against pathogens like influenza/Zika; (3) position circRNA as a transformative platform for pandemic response and global access. The modified sections (section 5, paragraph 2) are highlighted in the text.

Comments 10:The English language needs more improvement.

Response 10: We have carefully revised the language and corrected typographical errors throughout the manuscript.

Reviewer 3 Report

Comments and Suggestions for Authors

Yang-Yang Zhao et al. describe in their review article the role circular RNA in modern therapeutic regimes and vaccination strategies might play. 

The key features of circular RNA is that they can lead to protein expression and that they have a longer life time and a smaller immunogenicity than linear RNA when correctly designed. The protein can substitute missing internal proteins or lead to an immune reaction of the host. This immune reaction can have protective function against pathogens, help in fighting a growing tumor or suppress autoimmune reactions in autoimmune diseases when directed against key proteins that sustain the autoimmune process or by reprogramming T-cells. 

This is an excellent review article that gives a very good overview of the large and growing field of mRNA therapeutics. 

The Introduction section introduces the reader into the field by highlighting the recent break-throughs in the development of mRNA and circular RNA as therapeutics. 

- I would appreciate a small graphic showing the molecular mechanism how circular RNA is synthesised. This could motivate the emphasis the authors put on self-splicing RNAs. 

In the second chapter the authors explain the major advantage of circular RNA - its longer lifetime. Figure 1 gives an excellent overview of the various degradation pathways. None of them is based on the fastest - exconucleases facilitated degradation. 

Figure 2 describes in a similar, very transparent way the various ways the immune system can react to circular RNA introduced into the organism. The text describes this reaction in a more detailed way. 

After this introduction to the biology of circular RNA chapter 3 highlights the consequences these processes have for the optimal design of circular RNA to be highly efficient in protein expression. This is a very logical and well to understand sequence of information. 

Chapter 4 highlights that because of the complexity of all the factors discussed artificial intelligence has its place in considering all this information to propose better RNA sequences. 

Finally chapter 5 lists a series of successes circular RNA had in the laboratory and argues for the conclusion of this article that circular RNA has a very promising future in therapy and vaccination. 

I think this is an exemplary review article that guides the reader through the complexity of the field and never looses the reader because the article has a very logical structure and very good graphics and tables. 

The only proposal I have to make is that I would appreciate a small graphic describing the main mechanism of how circular RNAs are synthesised as described in lines 101 - 105. This explains to the novel reader why the introduction of self-splicing RNAs was so important. 

Author Response

Comments:The only proposal I have to make is that I would appreciate a small graphic describing the main mechanism of how circular RNAs are synthesised as described in lines 101 - 105. This explains to the novel reader why the introduction of self-splicing RNAs was so important.

Response:The figure illustrating in vivo and in vitro synthesis of circRNA has been added to Section 2.1. Additionally, we have briefly supplemented the description of in vitro synthesis methods in the main text and highlighted the revised content.

Reviewer 4 Report

Comments and Suggestions for Authors

The review of Zhao et al summarizes available data on unique features of circular RNA and potential applications of circRNA-based products.

The review is extensive and well written.

Given the rapid progress in this field and paucity of reviews related to circRNA technology, I believe that this review can be published in its present form  with minor improvements.

However, I think that this work would have much more value for the scientific community if authors would make an attempt to analyze in greater detail available non-clinical data on circRNA-based products in order to demonstrate how circRNAs  differ from mRNA products in terms of their safety and immunogenicity.

Minor issue: references to circRNA non-clinical studies in cancer models should be moved from section 5.4 to section 5.2

Author Response

Comments 1: However, I think that this work would have much more value for the scientific community if authors would make an attempt to analyze in greater detail available non-clinical data on circRNA-based products in order to demonstrate how circRNAs differ from mRNA products in terms of their safety and immunogenicity.

Response 1: Chapter 5 now includes specific examples with more detailed non-clinical data showing how circRNAs differ from mRNA products which are highlighted in the revised in the manuscript.

Comments 2: Minor issue: references to circRNA non-clinical studies in cancer models should be moved from section 5.4 to section 5.2.

Response 2: Sections 5.2 and 5.4 both address circRNA therapies in cancer models but through different mechanisms: Section 5.2 focuses on vaccine therapy, whereas Section 5.4 explores protein replacement therapy. The circRNA examples metioned in section 5.4 act as functional proteins such as signal transductors in cancer therapy instead of functioning as tumor antigens which triger immune response. Hence, we keep the original layout here.
